# Strategies to Improve Chimeric Antigen Receptor Therapies for Neuroblastoma

**DOI:** 10.3390/vaccines8040753

**Published:** 2020-12-11

**Authors:** Piamsiri Sawaisorn, Korakot Atjanasuppat, Usanarat Anurathapan, Somchai Chutipongtanate, Suradej Hongeng

**Affiliations:** 1Division of Hematology and Oncology, Department of Pediatrics, Faculty of Medicine Ramathibodi Hospital, Mahidol University, Bangkok 10400, Thailand; piamsiri.saw@mahidol.ac.th (P.S.); korakot.atj@mahidol.ac.th (K.A.); usanarat.anu@mahidol.ac.th (U.A.); 2Pediatric Translational Research Unit, Department of Pediatrics, Faculty of Medicine Ramathibodi Hospital, Mahidol University, Bangkok 10400, Thailand; 3Department of Clinical Epidemiology and Biostatistics, Faculty of Medicine Ramathibodi Hospital, Mahidol University, Bangkok 10400, Thailand; 4Chakri Naruebodindra Medical Institute, Faculty of Medicine Ramathibodi Hospital, Mahidol University, Samut Prakan 10540, Thailand

**Keywords:** CAR T cells, immunotherapy, pediatric neuroblastoma, strategy

## Abstract

Chimeric antigen receptors (CARs) are among the curative immunotherapeutic approaches that exploit the antigen specificity and cytotoxicity function of potent immune cells against cancers. Neuroblastomas, the most common extracranial pediatric solid tumors with diverse characteristics, could be a promising candidate for using CAR therapies. Several methods harness CAR-modified cells in neuroblastoma to increase therapeutic efficiency, although the assessment has been less successful. Regarding the improvement of CARs, various trials have been launched to overcome insufficient capacity. However, the reasons behind the inadequate response against neuroblastoma of CAR-modified cells are still not well understood. It is essential to update the present state of comprehension of CARs to improve the efficiency of CAR therapies. This review summarizes the crucial features of CARs and their design for neuroblastoma, discusses challenges that impact the outcomes of the immunotherapeutic competence, and focuses on devising strategies currently being investigated to improve the efficacy of CARs for neuroblastoma immunotherapy.

## 1. Introduction

Neuroblastoma, an extracranial solid tumor that initiates from the sympathetic nervous system’s neuroendocrine tissue, is one of the most common causes of death in pediatric cancers [1,2]. It is often diagnosed during the perinatal period, which accounts for 8% in patients under 15 years. This childhood neoplasm appears each year in more than 600 cases in the United States and 200 in Japan [3,4,5]. According to clinical presentation, neuroblastoma is an extremely variant characteristic tumor. It ranges from an adrenal mass tumor that regresses without treatment to a metastatic tumor that causes critical illness [6]. At present, while intensive therapies can be beneficial for patients with localized disease, these therapies have frequently not been useful against patients with high-risk disease (approximately 40% of cases associated with the extent of metastases and genetic factors) nor patients with relapse [7,8]. Hence, novel therapies based on immunotherapy were subsequently developed to improve survival for high-risk patients.

One such approach is treatment with anti-GD2 monoclonal antibody, which has already been assessed in a Phase III clinical trial. The use of this antibody-based therapy was compiled into a therapeutic protocol for high-risk neuroblastoma patients and revealed promising results [9,10]. This effectiveness has led to other immunotherapeutic approaches, even though their integration into conventional multimodality therapies requires further investigation.

After discovering that GD2, a disialoganglioside highly expressed in most neuroblastomas, is also targeted by T cells, cellular immunotherapies including genetic engineering of T lymphocytes to express anti-GD2 chimeric antigen receptors (CARs) have emerged and are now being studied. With the combination of antigen specificity and cytolytic capacity, anti-GD2 CAR T cells have demonstrated safety and antitumor efficacy in relapsed neuroblastoma patients [11,12]. Various preclinical studies have improved the antitumor effects, proliferation, and cytokine release of CAR T cells, and some approaches have reached clinical trials (Figure 1). Currently, the anti-GD2 CAR T cell approach might represent a potential therapeutic for pediatric neuroblastoma.

However, unlike the success of using CAR T cells in hematological malignancies [13,14,15], the efficacy of CAR T cells in neuroblastomas has been limited by several factors, including the tumor microenvironment, T cell exhaustion, and T cells’ persistence and potency, which may lead to therapeutic resistance [16,17,18]. Therefore, CAR T cell improvement as a feasible alternative to conventional therapies for patients with neuroblastoma is still a significant challenge to achieving immunotherapy.

In this review, we provide an updated summary of preclinical and clinical experience of CAR-based neuroblastoma therapies and discuss the improvements of CAR in different ways that could overcome clinical problems of applying this approach for treating neuroblastoma. Defining these strategies would suggest an attractive route of improving the potency of CAR immunotherapy.

## 2. CARs in Neuroblastoma

The knowledge of tumor immunology has been used in translating this comprehension into productive cancer therapies. A number of immunotherapy approaches represent a new borderline in treating cancers. One such strategy to overcome tolerance in cancer is to genetically engineer immune cells to express CAR. This concept of artificial antigen-specific receptors first originated in 1989–1993 [19,20]. By fusing an antibody-derived binding domain to T cell signaling domains, the CAR construct gains the tumor antigen specificity and the capacity to induce multiple signals in the response of immune cells. Over three decades, tremendous progress has been made and CARs were refined into the first, second, third, and currently fourth generations of their structure. Based on the potential of CAR T cells directed against the CD19 protein for treatment of hematologic malignancies shown in clinical trials, cancer immunotherapy was named the “Breakthrough of the Year” in 2013 by *Science* [21]. In addition, the use of anti-CD19 CAR T cells for relapsed/refractory acute lymphoblastic leukemia and in children and young adults was approved by the US Food and Drug Administration (FDA) in 2017 and two licensed products of CAR T cells including tisagenlecleucel and axicabtagene ciloleucel have been launched [22,23]. This success, therefore, has brought new insights for clinical translation in treating solid cancers.

CARs have been developed to fulfill the applicability of adoptive cellular immunotherapy for neuroblastoma in a major histocompatibility complex (MHC)-unrestricted manner in effector T cells. Effector immune cells, commonly T lymphocytes, have been genetically engineered to express an extracellular antigen-binding domain that is mostly a single-chain variable fragment (scFv) joined with a transmembrane domain and an intracellular signaling domain. The first-generation CARs were designed to have a single CD3-ζ intracellular signaling domain. The second- and third-generation CAR products were improved by adding one or two costimulatory endodomains to the CD3-ζ motif to achieve the optimal activation and survival of CAR cells. Current intracellular endodomains based on the costimulatory receptors include CD27, CD28, 41BB, ICOS, and OX40 [17,24]. Each of the CAR design components reflects the variations of therapeutics achievement, and novel CAR engineering has been developed for decades to broaden CAR therapeutics in solid tumors like neuroblastoma [25].

### 2.1. Summary of CAR Experience

Several CAR approaches in neuroblastoma have been developed according to discovered putative cancer antigens. There are some novel target antigens for CAR T cell therapy in neuroblastoma that have been investigated in the preclinical phase (Figure 2).

Anaplastic lymphoma kinase (ALK), an oncogene expressed in neuroblastoma cells, is associated with familial neuroblastoma cases [26,27]. Anti-ALK CAR has demonstrated its effectiveness against this neuroblastoma subtype in vitro and in vivo [28,29]. This line of research also suggested that antigen density must be considered to achieve CAR T cell potential. Another tyrosine kinase receptor that may be rendered an ideal target for CAR therapies is glypican 2 (GPC2). The high expression of GPC2 on the neuroblastoma cell surface brought promising clearance of disseminated neuroblastoma in the mouse model by anti-GPC2 CAR T cells [30]. B7H3 (CD276), a checkpoint molecule expressed in neuroblastomas, is another candidate for CAR therapies of neuroblastoma [31,32]. This attractive target brought useful immunotherapeutic strategies, including monoclonal antibodies and CARs targeting B7H3. Recently, the efficacy of anti-B7H3 CAR has been demonstrated in vivo [33,34]. Many target antigens that are specific to neuroblastoma cells have also been more characterized. Such antigens, including neural cell adhesion molecule (NCAM or CD56), New York esophageal squamous cell carcinoma 1 (NY-ESO1), and preferentially expressed antigen in melanoma (PRAME), were investigated both in vitro and in vivo for safety and efficacy, which gained attention for further development as CAR features [35,36,37,38].

To date, only CAR T cells targeting L1-CAM (CD171) and GD2 have reached the early phase of clinical trials (Table 1). L1-CAM, an adhesion molecule in the immunoglobulin superfamily, is another suitable target in neuroblastoma [39]. Because of the specificity of CE7, the monoclonal antibody that can bind to the L1-CAM epitope, the anti-L1-CAM CAR with the scFv from CE7 was generated. The first-generation anti-L1-CAM CARs’ efficacy and safety were investigated in patients with relapsed/refractory neuroblastoma in a Phase 1 clinical trial [12]. To augment the persistence of anti-L1-CAM CAR, second-generation CAR was generated using a 41BB costimulation domain, followed by third-generation CAR, including CD28 costimulation addition, which is currently being investigated in phase 1 clinical trials [40,41]. Until now, the most critical target antigen in neuroblastoma has been GD2, a disialoganglioside highly expressed on neuroblastoma tissue [42]. Owing to the presence of this antigen during chemotherapy and the success of anti-GD2 monoclonal antibody therapy, this antigen has been the most studied targeted for CAR T cell therapy in neuroblastoma [43]. Many approaches of first-generation anti-GD2 CAR have been reported, including anti-GD2 CAR containing a single-chain variable fragment (scFv) derived from 14g2a monoclonal antibody or Epstein–Barr virus-specific cytotoxic T cell transduced CARs (so-called GD2 CAR-CTL), with the knowledge that the prolonged persistence in vivo was associated with the costimulation domain of CAR [44,45,46,47]. Anti-GD2 CAR constructs are now considered on costimulatory endodomains. The second and third generations of CAR were then generated for in vitro and in vivo assessments of CAR T cell survival [48,49]. The third-generation anti-GD2 CAR, containing an inducible caspase 9 (iC9) safety switch, has been tested in clinical trials for its safety (clinicaltrials.gov identifier NCT01953900 and NCT01822652). Various clinical trials based on CAR therapy are underway to augment the reliable therapeutic outcomes. However, improving the efficacy and persistence of CAR is still a significant issue.

### 2.2. Obstacles to Using CARs in Neuroblastoma

#### 2.2.1. CAR T Cell Persistence and Exhaustion

Restrictive CAR T cell persistence has occurred as a major problem in neuroblastoma. Evidence from the first generation of CAR studies in vivo and clinical trials suggested that the limited persistence of CAR T cells from low activation and proliferation of cells also affected the antitumor efficacy [18,48,50,56]. One clinical study demonstrated that the infused, first-generation, anti-L1-CAM CAR cells were detectable in the peripheral blood up to 1–7 days after adoptive transfer in most patients with bulk disease but significantly longer (42 days) in a patient with limited disease burden [12]. T cell exhaustion might be a significant cause of shortening persistence. This is confirmed by discovering the exhausted CAR T cell phenotype in GD2 CAR T cells with low-level tonic signaling [57]. The persistence of infused CAR T cells might be prolonged if the exhaustion was reduced. Thus, several methods have been proposed to increase the persistence of CAR T cells. One such way is the utilization of second- and third-generation CARs, which improve costimulation after antigen binding (e.g., 4-1BB costimulatory domain) to protect shortened persistence. This development is under investigation for feasibility [13,15,57].

#### 2.2.2. Target Selection and On-Target, Off-Tumor Effect

Ideally, the target antigen for CAR T cells should have a high expression in cancer cells, a low expression in normal cells, and not be associated with oncogenesis [58]. It is known that there are challenges in the path of choosing an optimal CAR T cell target antigen in neuroblastoma, since many target antigens are related to normal peripheral nerves or neural tissue expression. Toxicities caused by particular interactions between the CAR and its target antigen expressed by normal cells termed on-target, off-tumor effects have been reported in previous CAR studies in solid tumors [59,60,61,62]. One clinical trial in metastatic colon cancer reported pulmonary infiltration by CAR T cells that caused a systemic cytokine storm in patients who received HER2-targeted CAR T cell therapy, demonstrating strong evidence of on-target, off-tumor toxicities [61]. On the other hand, there was no such effect in a pediatric sarcoma study using anti-HER2 CAR T cells and anti-GD2 CAR T cells in neuroblastoma studies [47,63,64]. This evidence suggested that the variation of antigen density on the different types of cancer is an additional factor to consider during target selection to avoid on-target, off-tumor effects.

#### 2.2.3. Tumor Microenvironment

Unlike the remarkable success of CAR T cells in the treatment of hematological malignancies, the efficacy of CAR T cells in neuroblastoma can be obstructed by the immunosuppressive tumor microenvironment (TME), which is a manifest barrier to achieve full effective CAR T cell therapy for solid tumors [65]. Significant factors derived from TME in neuroblastomas include immunosuppressive cells like tumor-associated macrophages (TAMs), Type 2 regulatory T cells (Tregs), and myeloid-derived suppressor cells (MDSCs), which contributed to poor results of CAR therapy [16]. Another factor is the inhibitory ligands present in the TME, such as PD-L1, the ligand for an inhibitory receptor expressed on activated T cells, named PD-1 [66,67]. Remarkably, this habitual expression of the inhibitory ligand in neuroblastoma can cause the loss of CAR T cells [68]. In addition to inhibitory signals, the availability of soluble factors in the TME, including galactin 1 and 3, TGF-β, and IL-10, can trigger T cell inhibitory pathways or inhibit T cell function [67,69,70,71,72], while secretory HMGB1 may be responsible for Treg differentiation in the neuroblastoma TME [73]. Furthermore, there are physical barriers that prevent the tumor access of T cells, such as protease fibroblast activation protein (FAP) expressed by tumor-associated stromal fibroblasts, the extracellular matrix (ECM), and immunosuppressive tumor vasculature-like vascular endothelial growth factor (VEGF) [74,75].

#### 2.2.4. CAR Trafficking

Trafficking of CAR cells into solid tumor sites to exert antitumor activity needs to be improved, especially in neuroblastoma. Several chemokines that can mediate immune cell trafficking are generally excreted by tumor or stromal cells like CC-chemokine ligand 17 (CCL17), CCL22, and CCL2 to enhance the localization of immune cells [76]. Moreover, suitable trafficking of immune cells, like T cells, can occur when there is an upregulation of a chemokine receptor that is matched to chemokine-related trafficking on T cells. However, in a previous study using CAR T cells derived from neuroblastoma patients, low expression of CCR2, a chemokine receptor, was detected [77,78,79]. Thus, various approaches to generate CAR T cells with an ability to traffic to neuroblastoma sites are underway.

## 3. Strategies to Improve CARs in Neuroblastoma

Remarkably the engineering of T lymphocytes to target tumor-associated antigens by forced expression of CARs has been successful against CD19^+^ leukemia [80,81,82]. Similar results have not yet been achieved against neuroblastoma [50,83,84]. The key factors that challenge the success of CAR immunotherapies are the immune effector cells, the design of the CAR construct, and the intrinsic tumor factors. The proposed strategies for improving CARs’ efficacy in neuroblastoma therapy are illustrated in Figure 3.

### 3.1. Improving Effector Immune Cells

Solving CAR T cell obstruction or the use of other effector cells might increase CARs’ tumor-killing activity in neuroblastoma. Regarding the killing effect against tumor cells, natural killer T (NKT) [85], gamma-delta T (γδT) [86], and natural killer (NK) cells [87] are valuable candidates for generating CAR-derived cells. These immune cells have different pros and cons. Thus, choosing the proper effector cell for each tumor target is extremely important. This part will assemble information about these CAR-derived effector cells, some of which are summarized in Table 2.

#### 3.1.1. T lymphocytes

To date, the immunophenotype of T cells is well understood as an essential parameter in safety and efficacy features in CAR T products [50,88]. The ratio of subtype composition and an available naïve and central memory T (T_CM_) cell population are essential to increase the therapeutic efficacy of CAR T cells. Shah et al., reported that enriching CD4^+^ and CD8^+^ in the starting T cell population before CAR-lentiviral transfection can enhance the efficacy of CAR T cells on neuroblastoma [89]. The 1:1 ratio of subtype composition between CD4^+^ and CD8^+^ possibly promotes high efficiency with more safety in patients [84]. Furthermore, another study exhibited an increased percentage of CD4^+^ T cells and CD45RO^+^CD62L^+^ T_CM_ in CAR treatment of neuroblastoma, shown to prolong the persistence of CAR T cells in clinical trials [50]. Interestingly, CAR T cells derived from naïve T cells and T_CM_ cells strongly proliferated ex vivo, resulting in 89% of the CAR T cell population [90]. Likewise, the procession of CD62L, a standard marker of memory cells, in the CAR population is more effective against both hematological malignancy and solid tumors and promoted longer persistence of peripheral NKT and T cells [91,92,93]. Because of the timing of infiltration into the tumor site, recognizing the tumor antigen, and performing their function, the memory subtype of CAR T cells is an exigency [94]. Selection of the T cell subtype population during CAR production or before administration to patients is recommended for improving the efficacy of CAR T cells against neuroblastomas. Nonetheless, CAR T cells are mostly detected in peripheral blood but have lower tumor infiltration than other effector cells such as NKT and NK cells. Thus, a combination of other infiltrated effector cells or ECM remodeling enzymes may enhance the tumor infiltration of T cells. In contrast, neuroblastoma-targeted CAR T cells may be more appropriate for eradicating the circulating neuroblastoma cells in patients’ blood and lymphatic system.

#### 3.1.2. Natural Killer Cells

Natural killer (NK) cell, lymphoid components of the innate immune system, can prevent pathogen invasion and kill tumor cells without requiring previous stimulation like T lymphocytes [95]. NK cells are mostly found in the peripheral blood, liver, spleen, and bone marrow, and a few are available in the lymphoid tissues [96,97]. NK cells can localize, migrate, and infiltrate into tumor sites to elicit MHC-unrestricted cytotoxicity and secrete proinflammatory cytokines and chemokines against solid tumors [97]. In addition, NK cells showed antitumor responses via the NKG2D and cytotoxic adapter molecule DNAX-activating Protein 10 (DAP10) to eliminate MDSCs, which then reversed the suppressive TME and attracted immune effectors infiltration into the tumor site [98]. According to these antitumor features, NK has been applied in various types of CARs immunotherapy against both hematological malignancies and solid tumors [99,100,101]. Of note, the irradiated NK cell line NK-92 has been adopted as the effector for various designs of CAR to overcome the difficulty of sorting and ex vivo expansion of primary NK cells [102,103].

Several studies reported that CAR-modified NK cells responded to solid tumors and hematological malignancies. CXCR1-expressed NKG2D-targeted CAR NK cells had enhanced tumor trafficking with preserved antitumor effects in ovarian cancer xenograft models [104], while anti-CD147 CAR NK exhibited potent cytotoxicity against hepatocellular carcinoma (HCC) cell lines in vitro and patient-derived HCC xenograft mouse models [100]. As aforementioned, the NK-92 cell line has been used as a CAR effector cell. Li et al. [105] demonstrated that bi-specific PD1-DAP10 and NKG2D CAR NK-92 cells enhanced cytotoxicity to human gastric cells (SGC7901), via triggering apoptosis. These dual-targeting CAR NK-92 cells also displayed significant antitumor activity in the SGC-7901 mouse xenograft. Moreover, the human prostate-specific membrane antigen (PSMA)-targeted modified CAR NK-92 cells recognized and mediated potent antitumor effects against prostate cancer xenograft models [106]. Anti-CD19 CAR NK cells showed potent tumor-specific lethality against NK-resistant lymphoma [107]. In a Phase I/II trial, Liu et al. [108] reported that 8 out of 11 (73%) patients with relapsed or refractory CD19-positive cancers (non-Hodgkin’s lymphoma or chronic lymphocytic leukemia (CLL)) had a response to anti-CD19 CAR NK without the development of major toxic effects. These infused CAR NK cells expanded and persisted in patients at low levels for at least 12 months [108].

For the neuroblastoma treatments, NK cells might requisite proper CAR constructs such as costimulators and/or signaling molecules, which are discussed in the next topic. Furthermore, several researchers indicated that the antitumor properties of CARs NK cells were downregulated by suppressive molecules of the TME, such as TGFβ, thereby limiting the tumor-killing capability of NK cells [98,109]. In this direction, a combination with the inhibitor of immunosuppressive molecules (e.g., anti-TGFβ) might increase the antitumor activities of neuroblastoma-targeted CAR NK cells.

#### 3.1.3. NKT Cells

NKT cells are a subset of innate lymphocytes that share T and NK cells’ properties [110,111]. NKTs have differences in T-cell receptor (TCR) diversity and their cancer immune response, and can be divided into two subtypes; invariant (Type 1, iNKT) and variant (Type 2) NKT cells [102]. Type I NKTs enhance antitumor immunity, while Type II NKTs regulate and inhibit tumor immunity [112,113,114]. iNKT cells have been studied and applied as potent tumor-specific CAR effectors against lymphomas [92,115] melanomas [116], and neuroblastomas [85]. The first use of CAR NKT cells against neuroblastoma was reported in 2014 [85]. Both the second and third generations of anti-GD2 CAR NKT cells exhibited potent antitumor activity and presented significantly prolonged survival without any graft-versus-host disease in a mouse xenograft model [85]. Unlike T cells, NKT effectively traffic and infiltrate into the tumor site [92] to mediate antitumor activities, inhibit tumor-supportive macrophages, and transactivate the localized NK and CD8^+^ T cells [117,118]. Additionally, a higher percentage of NKT cells in tumor-infiltrating lymphocytes have been detected in neuroblastoma than in patients’ peripheral blood [119]. Likewise, Xu et al. reported that CAR NKT cells enhanced CARs’ treatment efficacy through their persistence and provided high localization onto the tumor site without significant histopathological toxicity [52]. These suggested that NKT cells are more easily infiltrated into neuroblastomas than T lymphocytes. Recently, Heczey et al., [120] reported interim results from three patients with relapsed or resistant neuroblastoma, who enrolled into the Phase I dose escalation trial, receiving dose Level 1 of autologous NKT cells coexpressing GD2 CAR and interleukin-15 (NCT03294954). The results showed that GD2 CAR NKT cells expanded in vivo, localized to tumors, and, in one patient, induced regression of bone metastatic lesions without adverse effects [120].

Neuroblastoma-targeted CAR NKT cells are noticeable as potential effector cells against neuroblastomas and other solid tumors. Of note, an exhausting phenotype of NKTs upon in vitro stimulation is associated with the upregulated expression of PD-1 and TIM-3 [92]. Thus, CAR NKT cells should be used in combination with anti-PD-1 or anti-TIM-3 to generate an intense tumor-killing effect.

#### 3.1.4. γδT Cells

γδT cells are innate T lymphocytes that independently recognize MHC targets, have a natural killing activity against pathogens, and respond to tumor cells [121,122,123]. In contrast to αβT cells, γδT lymphocytes are rare in the peripheral blood and lymphoid organs (<10%), while they mostly reside in a wide variety of tissues including the skin epidermis, gastrointestinal tract mucosa, and the reproductive system [124,125]. Interestingly, γδT cells also found in neuroblastoma tumors [119]. γδT cells can be divided into three subpopulations depending on the differences in TCR-δ chain expression (Vδ1, Vδ2, and Vδ3), paired with the γ9 chain [126]. γδT cells expressing the Vγ9Vδ2 TCR is a common subset that can inhibit proliferation and angiogenesis and encourage tumor cell apoptosis [102]. Moreover, these γδT cells can be activated and proliferated ex vivo for more than 1000-fold expansion [127].

Neuroblastoma mostly escapes from the immune system via downregulation of the MHC Class I molecule [119]. Since γδT TCR does not require MHC-mediated antigen presentation [128,129], γδT cells may be appropriate to address this issue. Furthermore, these cells can differentiate into professional antigen-presenting cells (pAPCs), which present antigenic fragments for CD4^+^ and CD8^+^ αβT cells [130]. Although γδT cell exhaustion is associated with PD-1, an immune checkpoint molecule [131], γδT cells have lower expression of PD-1 compared with αβT cells, and are thereby recognized as a promising effector cell for CAR immunotherapy. Like T cells, the naïve and memory phenotype of γδT cells sufficiently survive to promote the persistence of CAR γδT cells in the recipients without decreasing the [132] killing activity against tumor cells [121].

Until now, few studies of CARs γδT cells have been reported. The first anti-CD19 CAR γδT cells showed strong killing efficacy against leukemia [122,132,133] For solid tumors, there have been preclinical studies of CAR γδT cells against melanoma [86] and neuroblastoma [127,134,135]. Therefore, γδT cells are influential candidates for developing CAR immunotherapy against neuroblastoma. At present, two Phase I clinical trials are available in clincaltrial.gov. The first study (NCT04107142) is planning to investigate of the safety and tolerability of NKG2DL-targeting CAR γδT cells in various solid tumors: colorectal, breast, sarcoma, nasopharyngeal carcinoma, prostate, and gastric cancer. The second clinical trial (NCT03885076) is estimating the potential role of anti-CD33-CD28 CAR γδT cells against acute myeloid leukemia (ALL). However, the clinical studies’ results are not yet available. From these properties, the γδT cell is a potential candidate for use as an effector immune cell in adoptive CAR immunotherapy against neuroblastoma.

### 3.2. Modification of CAR Constructs

The CAR construct design is one of the key features that affect the kinetics of expansion and the duration of persistence. CARs are recombinant receptors responsible for both antigen-binding and activation functions for immune cells [136]. The second-generation CARs offer a clinical benefit via both the TCR stimulatory domain (CD3z) and a single costimulatory domain [89]. Recently, FDA-approved products can contain either CD28 or 4-1BB (CD137) costimulatory domain [80]. Choosing the appropriate combination of costimulatory ligands, chimeric costimulatory receptors, or cytokines in the CARs’ structure for each objective immune cell may differently enhance the effector cells’ function, proliferation, and survival [136]. Good design of CAR structures can overcome the significant barriers of CARs in cancer immunotherapy.

#### 3.2.1. Hinge and Transmembrane Domain

The access of CARs to the target antigen depends on their flexibility and the length of the hinge and transmembrane domain [137]. In the case of anti-CD19 CAR T cells, human CD8α hinge and transmembrane molecules did not promote efficacies differently against tumor xenografts but induced lower cytokine levels compared with CD28 hinge and transmembranes [138,139]. These confirmed that the length and composition of the hinge provided an antigen binding and signaling cascade through the CAR T cells. Accordingly, the CD8α hinge and transmembrane domain may be an excellent choice to generate neuroblastoma-targeted CARs. However, further studies on this topic are required to discover a suitable composition that offers the highest tumor-killing efficacy but bestows the patients with low or no side effect.

#### 3.2.2. Costimulatory Domains

Costimulator molecules play a critical role in the development, activation, and functional response of effector immune cells [92]. A set of costimulatory receptors that are noticeable in human T cells include: CD28, 4-1BB, and OX40 [92], while the costimulators for activating NK cells are CD28, 4-1BB, 2B4, and NKG2D [140,141]. For creating CAR T cells, CD28 has more potent signaling through T cell activation with a short persistence, while 4-1BB provides a less potent signal but longer persistence [142]. To address cytokine releasing syndrome (CRS), the CD28 costimulator is appropriate for the dim target antigen, whereas 4-1BB is suitable for the high density target [139]. Unlike T cells, the CD28 costimulator is suitable for enhancing the proliferation, cancer-killing function, and persistence of NKT cells. Even though CD28 and 4-1BB prevent the loss of CD62L expression that is important to prolong the survival of NKT cells [92], the CD28 costimulator also forces the CAR NKT cell to massive expansion, while 4-1BB induces excessive activation, leading to cell death [52]. For CAR γδT cells, both CD28 and 4-1BB [143] were applied as costimulators [144,145]. However, data on the comparison and optimization of costimulators for CAR γδT cells are scanty. For NK cells, CD28 is mostly used as a costimulator. Oelsner et al. suggested that CD19-CAR NK cells with CD28 costimulators showed better performance against B-cell malignancy than 4-1BB [145]. Fusing DNAM1 and 2B4 costimulatory domains in CAR NK design can enhance cytotoxicity against hepatocellular cancer cells in vitro [146].

#### 3.2.3. Signaling-Transducing Domain

CD3ζ is commonly used as the signaling molecule in CAR constructs to activate T and NK cells functions for both hematologic malignancy and solid tumors [81,89,138,147]. Immunotherapy resistance and CRS are associated with highly active CD3ζ-activated CAR T cells [148,149]. Implementing other activation domains in CAR designs should be considered as an alternative to provide a robust clinical advantage in relieving the risk of CRS and overcoming these barriers.

DNAX activation protein of 12 kDa (DAP12), an immunotyrosine-based activation motif-containing adaptor, is constitutively expressed in NK cells and is expressed in a subset of human T cells [150]. The first use of DAP12 as an alternative cytoplasmic domain in CAR design emerged in 2015 [151]. The chimeric target receptor fusing with the transmembrane and cytoplasmic domains of KIR2DS2, a stimulatory killer immunoglobulin-like receptor, and DAP12 (KIR-CAR DAP12) triggered robust antigen-specific proliferation, effector function, and enhanced antitumor activity in leukemia cell lines [151]. Although only a few studies have used DAP12 as a new activation domain in CAR-modified T cells, the low cytokine markers of CRS could be observed in DAP12-based CAR preclinical studies [148]. Compared with CD3ζ-based second-generation CARs, T cells modified with the natural killer group 2D (NKG2D) ectodomain combined with 4-1BB and the DAP12 signaling domain released lower levels of interferon-gamma (IFN-γ), tumor necrosis factor-alpha (TNF-α), and interleukin-2 (IL-2) during tumor cell lysis without a significant difference in tumor-killing effects both in vitro and in vivo [148]. DAP12 is also involved in the signal transduction of activating NK cells [152]. DAP12-based CAR NK cells could eradicate neuroblastomas in the mouse xenograft model [152]. This DAP12 might be a brilliant new signaling molecule for high efficacy with more safety in CAR development

#### 3.2.4. Other Generations of CAR Structures

Although the second generation of CARs was mostly used in neuroblastoma clinical trials, the fourth generation of CARs has been developed. With neuroblastoma-targeted antigens incorporating multiple costimulatory molecules and a suicide gene-inducible caspase9 (iCasp9), the success of fourth-CAR T cells against MYCN amplification in neuroblastomas has been reported [53]. This fourth-generation CAR exhibited higher tumor-killing activity than the third-generation CAR with a safety profile, leading to 4 years of survival of a neuroblastoma patient after fourth-generation CAR treatment [53]. Successful fourth-generation CAR T cells are being explored for the new era of treatment for neuroblastoma and other solid tumors.

### 3.3. Overcoming the TME to Improve the Efficacy of CAR Immunotherapy

The intrinsic tumor factors, including the extracellular matrix (ECM) and the TME, are critical challenges for CAR immunotherapy against neuroblastoma [153]. The counterstrategies for these tumor escape mechanisms are shown in this topic.

#### 3.3.1. ECM: Trafficking and Infiltration of Effector Cells

Low T cell infiltration has been observed in so-called immunologically “cold” neuroblastoma tumors [154]. Various studies demonstrated that poor trafficking to and limited CAR T cell persistence in solid tumors are significant burdens in CAR immunotherapy [155]. To increase the trafficking ability of CAR T cells via presenting matched chemokine receptors or adding therapeutic agents against tumor-enriched chemokines may be a practical strategy for neuroblastoma and other solid tumor treatments [156]. For example, in hepatocellular carcinoma in which a high expression level of CXCR2 ligand was observed, Liu et al. confirmed that using CXCR2 CAR T cells could enhance in vivo trafficking and tumor cytotoxicity [157]. Additionally, the fusion of IL-7 and CCL19 on the CAR structure showed increased infiltration and exhibited high antitumor activity of CAR T cells [158]. Moreover, given the prevalence of IL-8 production in human cancer [159], this approach may find broad applicability in the potentiation of CAR T cell immunotherapy for solid tumors. Accordingly, the tumor chemokine–chemokine receptor network in neuroblastoma should be identified. Studies showed that VEGF, IL-6, and IL-10 receptor (IL-10R) in neuroblastoma were associated with poor outcomes [160,161,162]. Thus, the fusion of anti-VEGF, anti-IL-6, or anti-IL-10R into the CAR structure may also be effective against neuroblastoma.

#### 3.3.2. Myeloid-Derived Suppressor Cells

MDSCs are immunosuppressive immune cells found in the TME [163,164]. These MDSCs enhance tumor growth and suppress the infiltration, proliferation, and tumor-killing activity of CAR T cells [165]. These cells also express PD-L1 themselves, which contributes to immune escape in various solid tumors [166]. Researchers found that the NKG2D ligand, a cytotoxicity receptor activated by non-classical MHC molecules, is overexpressed in these tumor-infiltrating MDSCs [167,168]. A combination with NKG2D-CAR NK cells enhanced the tumor infiltration and expansion of CAR T cells at tumor sites and prolonged CAR T cell survival compared with single treatment [98,169]. The mixed NKG2D-CAR NK and neuroblastoma-targeted CAR T cell treatment may safely enhance antitumor activity against neuroblastomas that are supported and protected by MDSCs.

#### 3.3.3. Tumor Extracellular Vesicles

Extracellular vesicles (EVs) are lipid bilayer vesicles released from the cells [170], which can be classified into three major groups (exosomes, microvesicles, and apoptotic bodies), based on their size [171]. Like cargo, EVs contain soluble proteins such as growth factors, cytokines, chemokines [172], lipids, metabolites, and nucleic acids, including regulatory microRNAs (miRs) [171,173]. Several studies reported that EVs play a crucial role in tumor–tumor [174,175] and tumor–immune cell communications [176] and affect tumor cells’ phenotype and metastatic potential [177,178,179]. Neuroblastoma cells release EVs in their extracellular space [172]. Previous studies showed that neuroblastoma EVs induced the production of pro-tumorigenic cytokines and chemokines such as IL-6, IL-8, VEGF, and CCL2 by MSCs [180]. Additionally, neuroblastoma EVs also trigger a proinflammatory response in monocytes and promote neuroblastoma chemoresistance [181]. Future studies aiming to reveal the exact roles of EVs in the communication between the neuroblastoma and the TME is required to decipher a novel treatment strategy against neuroblastoma TMEs.

### 3.4. Combination of CARs

#### 3.4.1. PD-1/PDL-1

Programmed cell death 1 (PD-1 or PDCD1) is a coinhibitory receptor expressed by all CAR effector cells, including αβT [182], γδT [143], NKT [52], and NK cells [131]. PD-1 binds to its ligand, PD-L1 [or CD274], leading to a decrease in effector cell receptors’ downstream signaling and diminishing T cell activation [183]. By using immunohistochemical analysis, Zuo et al. [184] reported that up to one-third (11/31) of neuroblastoma patients showed positive PD-L1 expression in tumor tissues. It was not surprising that PD-L1 positivity was associated with decreased overall survival [184]. PD-L1 expression in metastatic neuroblastomas also plays a key role in immune resistance mechanisms [185]. Several reports suggested that inhibition of PD-1/PD-L1 interaction by anti-PD-1 or anti-PD-L1 antibody enhanced T cells’ killing efficacy against tumor cells [67,186,187]. Li et al. reported that combining anti-PD-1 antibody with CAR T cells engineered to secrete PD-1 inhibitors can enhance expansion and inhibitory effects in a human lung carcinoma xenograft mouse model [188]. The modified human CAR T cell secreting PD-1 blocking scFvs (E27-secreting CAR T) had also been established and investigated for therapeutic efficacy in hematopoietic malignancies as compared with their prototype CAR T cells. The results showed that E27-secreting CAR T cells significantly eliminated tumor cells and increased the survival of tumor-bearing mice [189]. Rupp et al. successfully established PD-1 deficient anti-CD19 CAR T cells combined with Cas9 ribonucleoprotein (Cas9 RNP)-mediated gene editing to disturb PD-L1-positive tumor xenografts. The results showed that Cas9-edited CAR T cells can improve the therapeutic efficacy for cancer immunotherapy in vivo [190]. Since PD-L1 is highly upregulated in multiple solid tumors, PD-L1-targeting CAR T cells have been designed. A vector bearing the extracellular and transmembrane regions of human PD-1 (dPD1z) and a CAR vector against PD-L1 (CARPD-L1z) have been established. Both dPD1z and CARPD-L1z effectively suppress the growth of multiple types of tumors in patient-derived xenograft models [191].

The combination of anti-PD1 and CAR T therapy has been investigated in Phase I clinical trials for hematological (NCT04134325, NCT03287817, NCT04337606) and malignant pleural mesothelioma (NCT04577326). Likewise, the safety and efficacy of anti-PD-L1 combined with CAR T cells have been investigated for B-cell malignancies (NCT03310619) and various types of solid tumor including cervical cancer, sarcoma, and non-small cell lung cancer (NCT04556669). In this direction, checkpoint inhibition may be turning the neuroblastoma microenvironment to respond to immunological treatment and enhance the efficacy of CAR therapy against neuroblastomas as well [192,193].

#### 3.4.2. Cytokines

Another approach to improve the potency of CARs is to genetically modify the effector cells secreting pro-inflammatory or pro-proliferative cytokines, aiming to strengthen effector function, proliferation, and persistence, and to alter the TME [136,194]. The cytokines IFN-γ, IL-2, IL-7, IL-12, IL-15, IL-18, IL-21, and IL-27, promoted neuroblastoma regression via enhancing the efficacy of effector cells, whereas VEGF, IL-6, and IL-10 induced neuroblastoma progression through their immunosuppressive activities. Several studies reported that increasing the proinflammatory chemokine IL-8 levels in patients was related to poor prognosis in neuroblastomas and other solid tumors [195,196]. This information is useful for the new design of CAR constructs or combination therapy to enhance the efficacy of CARs against neuroblastoma.

To date, IL-15 cytokine has been studied for its ability to enhance CARs’ antitumor effects. IL-15 plays a crucial role in T and NKT cells [52,197] via enhanced CAR antitumor activity and survival in peripheral blood and tumor tissue [197]. This study suggested that the IL-15-conjugated CD28-CAR structure reduced the exhaustion marker and improved the persistence and the tumor-killing activity of NKT-established CARs. Recently, Xin et al. utilized mice bearing neuroblastoma xenografts to demonstrate that GD2-CAR IL-15 NKT cells enhanced in vivo persistence, increased localization to tumor sites, and improved tumor control as compared with GD2-CAR NKT cells [52]. Interestingly, IL-15-conjugated GD2 CAR T cells reduced the expression of the PD-1 receptor [197]. These characteristics have made IL-15 a promising candidate for CAR T (or NKT) design for neuroblastoma immunotherapy.

## 4. Conclusions and Future Perspectives

The strategies to improve CARs for use in neuroblastomas are mostly concerned with increasing the antitumor activity and persistence of the infused CAR cells. Therefore, discovering new tumor antigens that are more restricted to neuroblastomas with less neuronal toxicity is very necessary. The evaluation of new target antigens such as B7H3 or o-acetyl-GD2 (oaGD2) is ongoing [198,199]. Researchers can also modify the construct of CARs to be more effective by adding new costimulatory endodomains such as 4-1BB or inducible T-cell costimulator (ICOS), which can be seen in the fourth generation of CAR therapy [200,201]. Seeking a way to overcome the immunosuppressive TME in neuroblastomas is a critical challenge in CAR studies. Various trials of CAR cell therapy in neuroblastomas, which relied on the strategies mentioned above, are underway to validate improved outcomes. Alternatively, administration of antitumor-related cytokines might enhance the immune response and amplify the direct neuroblastoma cell-killing effect of CAR cells in a complex TME. For example, IL-15 has been used to improve in vivo persistence and antitumor activity against neuroblastoma. Hence, another area of interest in CAR structure design is the induction of cytokine expression.

Given all these findings, understanding the mechanism of how the TME affects the resistance of CAR cells and inhibits the in vivo antitumor function is vastly relevant. The exploration of exosomal miRs or exosome decoys released within the neuroblastoma TME may be essential alternatives to overcome the barriers posed by the TME. Furthermore, by understanding the molecular mechanisms within the TME, researchers can efficiently engineer molecular targeting CAR cells to restore TME resistance sensitivity.

Lastly, a more practical approach for CAR-modified cell manufacturing is still needed. The ability to manufacture prompt quantities of an efficacious cell product is required for neuroblastoma patients with various conditions. In the end, good manufacturing might help to create highly active CAR cells for patients with relapsed disease in time.

In summary, immunotherapy using CARs in neuroblastoma treatment has shown promising efficacy and safety. To utilize the therapeutic benefits of CARs as the first line of immunotherapy in patients with a high risk of relapsed neuroblastoma, several approaches have been explored further to augment the antitumor function of CAR-modified cells in vivo. However, unlike the remarkable success in hematological malignancies, various barriers restrict the effective use of this CAR treatment in clinical trials, as indicated earlier. The strategies provided in this review may offer suitable approaches to address the benefits of CAR therapy in neuroblastoma.

## Figures and Tables

**Figure 1 vaccines-08-00753-f001:**
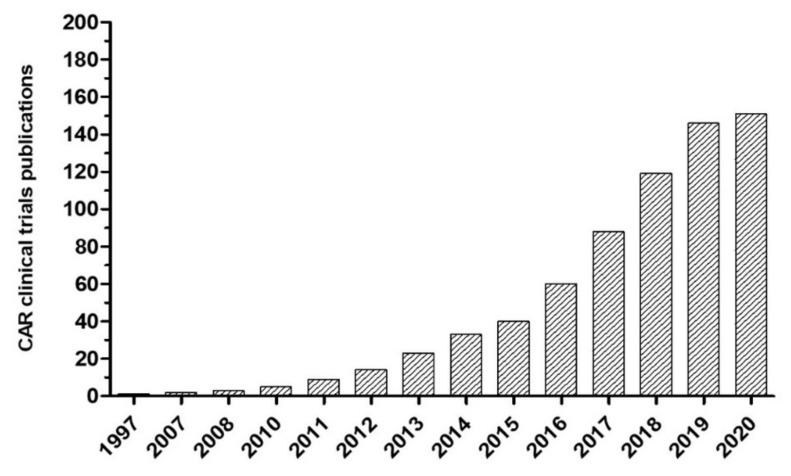
Cumulative publications of chimeric antigen receptor (CAR) immunotherapy in clinical trials.

**Figure 2 vaccines-08-00753-f002:**
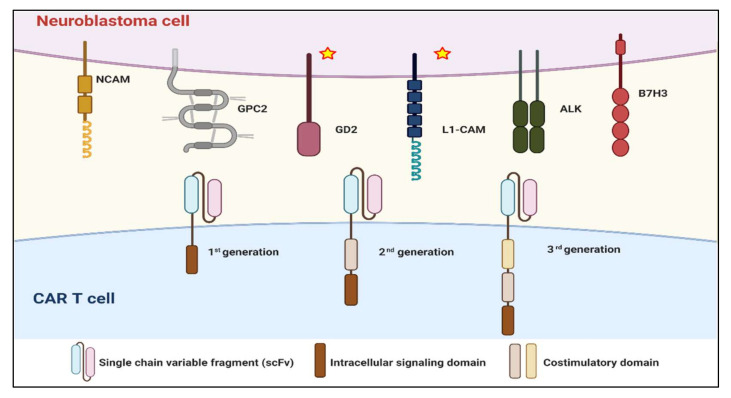
Target antigens conducted on the safety and efficacy of CAR therapy for neuroblastoma. Six surface antigens of neuroblastoma, including L1-CAM, GPC2, NCAM, GD2, ALK, and B7H3, are under development and investigation. L1-CAM and GD2 are the only two target antigens currently in completed clinical trials for neuroblastoma (labeled with star).

**Figure 3 vaccines-08-00753-f003:**
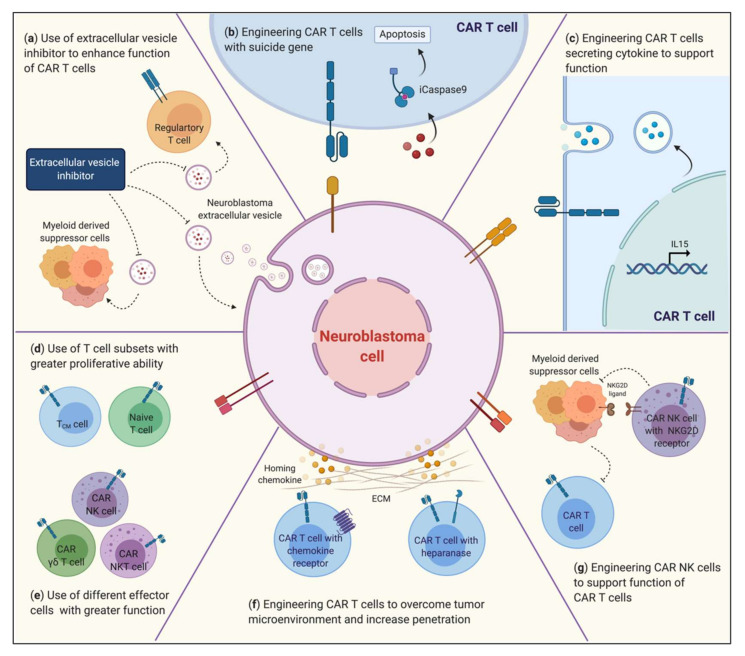
Proposed strategies to improve CARs’ efficacy against neuroblastoma. To enhance the antitumor toxicity of chimeric antigen receptor (CAR) T cells, several strategies can be applied: (**a**) The use of extracellular vesicle (EV) inhibitors. EVs are cargo ships containing functional molecules that enhance tumor growth and metastasis and may decoy CAR immunotherapy. Inhibiting the secretion of EVs from tumor cells may improve the killing efficiency CAR T treatment. (**b**) Engineering CAR T cells with a suicide gene. To improve CARs, and any side effects of CARs, fusing the iCaspase-9 gene in CARs’ structure can break T cell activation via induction of apoptosis in CAR T cells; (**c**) Engineering CAR T cells secreting cytokine to support function. IL-15 cytokines showed an ability to increase CARs’ antitumor toxicity in both T and natural killer T (NKT) cells. (**d**) Use of T cell subsets with greater proliferative ability. Naïve and central memory effector cells have the potential to enhance both CARs’ efficacy and the persistency of immune effector cells. (**e**) Use of different effector cells with greater function. For various solid tumors, CARs derived from natural killer (NK), NKT, or γδT cells showed more penetration into tumor sites and more potent toxicity than CAR T cells. (**f**) Engineering CAR T cells to overcome the tumor microenvironment (TME) and increase penetration. Supplementing with tumor-enriched chemokine inhibitor or using extracellular matrix remodeling enzyme during treatment might improve the trafficking and infiltration of CAR T cells; (**g**) Finally, engineering CAR NK cells to support the function of CAR T cells. Inhibition of immunosuppressive cell function in the TME via CAR NK cells can enhance the homing and efficacy of neuroblastoma-targeted CAR T cells.

**Table 1 vaccines-08-00753-t001:** An outline of clinical trials for CAR immunotherapy in neuroblastoma.

Clinicaltrials.Gov Identifier	Type of Study	Status	Title	Interventions	Target/ScFv/Signaling Domain	Date	Key Result	Reference
N/A	Phase I *N* = 6	Completed	N/A	Anti- L1-CAM CAR T cells	L1-CAM/CE7R/ CD3ζ	N/A	Partial response in one patient	[12]
NCT02761915	Phase I *N* = 27	Recruiting	A UK cancer research trial of anti-GD2 T cells (1RG-CART)	Anti-GD2 CAR T cells	GD2/KM8138/ CD28.CD3ζ	Study start: February 2016 Primary completion: February 2023	On target activity in bone and bone marrow were detected	[47]
NCT01822652	Phase I *N* = 11	Active, not recruiting	Third generation GD-2 chimeric antigen receptor and icaspase suicide safety switch, neuroblastoma, GRAIN	iC9-GD2 CAR T cells	GD2/14g2a/ CD28.OX40.CD3ζ	Study start: August 2013 Primary completion: December 2015	Modest early antitumor responses were detected	[49]
NCT00085930	Phase I *N* = 19	Active, not recruiting	Blood T-cells and EBV specific CTLs expressing GD-2 specific chimeric T Cell receptors to neuroblastoma patients	EBV-specific CTLs	GD2/14g2a/ CD3ζ	Study start: April 2003 Primary completion: January 2010	Complete response in one patient	[18,50]
NCT01460901	Phase I *N* = 5	Completed	Study of donor-derived, multi-virus-specific, cytotoxic T-lymphocytes for relapsed/refractory neuroblastoma (STALLONe)	GD2 CAR modified Tri-virus-specific cytotoxic t-cells	GD2/14g2a/CD3ζ	Study start: October 2012 Primary completion: January 2015	Partial response in 30% of patients	[51]
NCT03294954	Phase I *N* = 24	Recruiting	GD2-specific CAR and interleukin-15 expressing autologous NKT cells to treat children with neuroblastoma (GINAKIT2)	GINAKIT Cells	GD2/14g2a/ CD28.CD3ζ	Study start: 18 January 2018 Primary completion: 1 September 2021	N/A	[52]
NCT02765243	Phase II *N* = 34	Recruiting	Anti-GD2 fourth generation CART cells targeting refractory and/ or recurrent neuroblastoma	Anti-GD2 CAR T cells	GD2/N/A/CD28.4-1BB.CD27.CD3ζ	Study start: 23 March 2016 Primary completion: 12 May 2019	Partial response in 15% of patients	[53]
NCT03618381	Phase I *N* = 36	Recruiting	EGFR806 CAR T cell immunotherapy for recurrent/refractory solid tumors in children and young adults	Anti-EGFR806-EGFRt CAR T cells	EGFR/N/A/4-1BB.CD3ζ	Study start: 18 June 2019 Primary completion: June 2021	N/A	[54]
NCT01953900	Phase I *N* = 26	Active, not recruiting	iC9-GD2-CAR-VZV-CTLs/refractory or metastatic GD2-positive sarcoma and neuroblastoma	Anti-GD2 CAR T cells	GD2/14g2a/ CD28.OX40.CD3ζ	Study start: April 2014 Primary completion: April 2021	N/A	[54,55]
NCT03635632	Phase I *N* = 94	Recruiting	C7R-GD2.CAR T cells for patients with relapsed or refractory neuroblastoma and other GD2 Positive cancers (GAIL-N)	C7R-GD2 CAR T cells	GD2/14g2a/ CD28.OX40.CD3ζ	Study start: 23 April 2019 Primary completion: June 2022	N/A	Unpublished
NCT03373097	Phase I/II *N* = 42	Recruiting	Anti-GD2 CAR T cells in pediatric patients affected by high risk and/or relapsed/refractory neuroblastoma or other GD2-positive solid tumors	Anti-GD2 CAR T cells	GD2/14g2a/ CD28.4-1BB.CD3ζ	Study start: January 2018 Primary completion: December 2024	N/A	Unpublished
NCT03721068	Phase I *N* = 18	Recruiting	Study of CAR T cells targeting the GD2 with IL-15+icaspase9 for relapsed/refractory neuroblastoma	iC9-GD2 CAR T cells	GD2/N/A	Study start: 19 February 2019 Primary completion: 19 June 2024	N/A	Unpublished
NCT04637503	Phase I/II *N* = 100	Recruiting	4SCAR-T therapy targeting GD2, PSMA and CD276 for treating neuroblastoma	4SCAR T cell	GD2, PSMA, CD276/N/A	Study start: 18 November 2020 Primary completion: 19 November 2020	N/A	Unpublished
NCT04539366	Phase I *N* = 67	Not yet recruiting	Testing a new immune cell therapy, GD2-targeted modified T-cells (GD2CART), in children, adolescents, and young adults with relapsed/refractory osteosarcoma and neuroblastoma, the GD2-CAR persistence trial	Anti-GD2 CAR T cells	GD2/N/A/ CD28.OX40.CD3ζ	Study start: 1 January 2021 Primary completion: 1 August 2024	N/A	Unpublished
NCT04483778	Phase I *N* = 68	Recruiting	B7H3 CAR T cell immunotherapy for recurrent/refractory solid tumors in children and young adults	Anti B7H3-EGFRt-DHFR CAR T cells	B7H3/N/A/4-1BB.CD3ζ	Study start: 13 July 2020 Primary completion: December 2025	N/A	Unpublished
NCT02107963	Phase I *N* = 15	Completed	A Phase I trial of T cells expressing an anti-GD2 chimeric antigen receptor in children and young adults with GD2^+^ solid tumors	Anti-GD2 CAR T cells	GD2/14g2a/OX40.CD28.CD3ζ	Study start: 28 February 2014 Primary completion: 15 August 2016	N/A	Unpublished

**Table 2 vaccines-08-00753-t002:** Comparison of CAR-derived effector cells for neuroblastoma.

Effector Cell	Ex vivo Expansion	Type of Target Cells		CAR Design		Combination		
		Localized	CTC	Costimulating	Signaling	Cytokine	Effector Cell	Signaling Inhibitor
T	***	*	***	Dim expression target: CD28 High expression target: 4-1BB	Normally: CD3ζ Decrease CRS: DAP12	IL-2, IL-7, IL-15	NKG2D.CAR NK	anti-PD-1/PD-L1, anti-IL-10, IL-6 inhibitor
NK	*	***	**	CD28, 4-1BB, DNAM1 and 2B4	CD3ζ and DAP12	IL-2, IL-7, IL-21	NB-targeted CAR T	anti-PD-1/PD-L1, anti-TGFβ, anti-IL-10, IL-6 inhibitor
NKT	**	***	*	CD28	CD3ζ and DAP12	IL-2, IL-7, IL-15	N/A	anti-PD-1/PD-L1, anti-IL-10, IL-6 inhibitor
γδT	***	***	*	Both CD28 and 4-1BB	CD3ζ [no data for DAP12]	IL-2, IL-7, IL-15	N/A	anti-PD-1/PD-L1, anti-IL-10, IL-6 inhibitor

Score: *** = high, ** = moderate, * = low. CAR, chimeric antigen receptor; CTC, circulating tumor cells; DAP12, DNAX-activating protein of 12 kDa; DNAM1, DNAX accessory Molecule-1; DNAX-activating protein 12; ECM, extracellular matrix; IL, interleukin; N/A, not applicable; NB, neuroblastoma; NK, natural killer cell; NKG2D, natural killer cell receptor D; NKT, natural killer T lymphocyte cell; PD-1, programmed cell death-1; PD-L1, programmed cell death-ligand 1; T, T lymphocyte; γδT, gamma delta T lymphocyte.

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
