# Peer review of "Strategies to Improve Chimeric Antigen Receptor Therapies for Neuroblastoma"

_vaccines, 2020, doi:10.3390/vaccines8040753_

Round 1

Reviewer 1 Report

In this review, the authors summarized the preclinical and clinical experience of CAR-based neuroblastoma therapies, and the improvements of CAR in different ways were further discussed, which could provide reference for overcoming the clinical problems of treating neuroblastoma by the application of this approach. The strategies supplied in this review may suggest an attractive route of improving the potency of CAR immunotherapy. Generally, this review is representative and referable. However, attentive proofreading and modification are needed before consideration for publication.

  1. Please supplement and enrich the development process of CARs and some important progress events.
  2. Some novel target antigens for CAR T cell therapy in neuroblastoma were illustrated in Figure 2, which have been investigated in the preclinical phase, and the proposed strategies for improving CARs efficacy in neuroblastoma therapy were introduced in Figure 3. It is better for the author to beautify Figure 2 and Figure 3 to make the picture more convincing and vivid.
  3. Valuable candidates for generating CAR-derived cell including the killing effect against tumor cells, natural killer T (NKT), gamma-delta T (γδT), and natural killer (NK) were introduced in the manuscript, respectively. It is better to elaborate and expand on the typical applications of the above cells in CAR derivation.
  4. When introducing the combination of PD-1/PDL-1 and CARs, it is suggested that the author give some specific examples in the literature or in clinical experience of CAR-based neuroblastoma therapies.
  5. It is suggested that the author integrate the “Future perspective” of the fourth part and the “Conclusions” of the fifth part into one part.
  6. Some typos needed to be corrected, eg: “… This 1st generation anti-L1-CAM CAR efficacy and safety was investigated in patients with relapsed/refractory neuroblastoma in phase 1 clinical trial”, “It is known that there are challenges in the way of choosing an optimal CAR T cell target antigen in neuroblastoma is challenging …”, “2.3.4. CAR trafficking”.
  7. The reference format needs to be unified according to the requirements of the journal.

Author Response

Reviewer#1

Comments and Suggestions for Authors

In this review, the authors summarized the preclinical and clinical experience of CAR-based neuroblastoma therapies, and the improvements of CAR in different ways were further discussed, which could provide reference for overcoming the clinical problems of treating neuroblastoma by the application of this approach. The strategies supplied in this review may suggest an attractive route of improving the potency of CAR immunotherapy. Generally, this review is representative and referable. However, attentive proofreading and modification are needed before consideration for publication.

The authors are very grateful to the reviewer for great suggestions and efforts towards improving the quality of our manuscript.  Please find below details of how all comments have been fully addressed in the manuscript.

1. Please supplement and enrich the development process of CARs and some important progress events.

RESPONSE: We thank the Reviewer for this suggestion. A paragraph described the development process of CARs and some important progress events was added into the ‘CARs in neuroblastoma’ section as following;

Page 2; line 69-83:

The knowledge of tumor immunology has been made in translating this comprehension into productive cancer therapies. A number of immunotherapy approaches represent a new borderline in treating cancers. One such strategy to overcome tolerance in cancer is to genetically engineer immune cells to express CAR. This concept of artificial antigen specific receptors was first originated in 1989–1993 [19,20]. By fusing an antibody derived binding domain to T cell signaling domains, the CAR construct gains the tumor antigen specificity and the capacity to induce multiple signals in response of immune cells. Over three decades, tremendous progress has been made and CARs were sophisticated as 1st, 2nd, 3rd, and currently 4th generation upon their structure. Based on the potential of CAR T cells directed against the CD19 protein for treatment of hematologic malignancies shown in clinical trials, cancer immunotherapy was named as “Breakthrough of the Year” in 2013 by Science [21]. In addition, the use of anti-CD19 CAR T cells for relapsed/refractory acute lymphoblastic leukemia and in children and young adults was approved by US Food and Drug Administration (FDA) in 2017 and licensed two products of CAR T cells including tisagenlecleucel and axicabtagene ciloleucel have been launched [22,23]. This successful, therefore, have brought the new insight for clinical translation in treating solid cancers.

2. Some novel target antigens for CAR T cell therapy in neuroblastoma were illustrated in Figure 2, which have been investigated in the preclinical phase, and the proposed strategies for improving CARs efficacy in neuroblastoma therapy were introduced in Figure 3. It is better for the author to beautify Figure 2 and Figure 3 to make the picture more convincing and vivid.

RESPONSE:  As the Reviewer’s suggestion, Figure 2 and Figure 3 were beautified to make the pictures more convincing and vivid.

3. Valuable candidates for generating CAR-derived cell including the killing effect against tumor cells, natural killer T (NKT), gamma-delta T (γδT), and natural killer (NK) were introduced in the manuscript, respectively. It is better to elaborate and expand on the typical applications of the above cells in CAR derivation.

RESPONSE: This is a constructive comment that we feel help improving the quality of our manuscript significantly. As the reviewer’s suggestion, paragraphs described the typical applications of CAR-derivednatural killer (NK), natural killer T (NKT), and gamma-delta T (γδT) were added as following;

Page 9; line 259-351:

“3.1.2. Natural killer cells

Natural killer (NK) cell, a lymphoid component of the innate immune system that can prevent pathogen invasion and kill tumor cells without requiring previous stimulation like T lymphocytes [96]. NK cells are mostly found in peripheral blood, liver, spleen, bone marrow, and a few are available in the lymphoid tissues [97,98]. NK cells can localize, migrate, and infiltrate into tumor sites to elicit MHC-unrestricted cytotoxicity and secretedproinflammatory cytokines and chemokines against solid tumors [98]. In addition, NK cell showed the antitumor responses via NKG2D cytotoxic adapter molecule DAP10 to eliminate MDSCs, which then reversed the suppressive TME and attracted immune effectors infiltration into tumor site [99]. According to these anti-tumor features, NK has been applied in various type of CARs immunotherapy against both hematological malignancy and solid tumors [100-102]. Of note, the irradiated NK cell line, NK-92, has been adopted as the effectors for various designs of CARs to overcome the difficulty of sorting and ex vivoexpansion of primary NK cells [103,104].

Several studies reported that CAR-modified NK cells responded to solid tumors and hematological malignancies. CXCR1-expressed NKG2D-targeted CAR NK cells had enhanced tumor trafficking with preserved anti-tumor effect in the ovarian cancer xenograft models [105], while anti-CD147 CAR NK exhibited potent cytotoxicity against hepatocellular carcinoma (HCC) cell lines in vitroand patient-derived HCC xenograft mouse models [101]. As aforementioned, NK-92 cell line has been used as a CAR effector cell. Li M et al., [106]demonstrated that bi-specific PD1-DAP10 and NKG2D CAR NK-92 cells enhanced cytotoxicity to human gastric cells; SGC7901, via triggering apoptosis. These dual targeting CAR NK-92 also displayed significant antitumor activity in the SGC-7901 mouse xenograft. Moreover, the human prostate-specific membrane antigen (PSMA)-targeted CAR modified NK-92 cells recognized and mediated potent anti-tumor effect against prostate cancer- xenograft models [107]. Anti-CD19 CAR NK showed potent tumor-specific killing against NK-resistant lymphoma [108]. In the phase I/II trial,Liu E, et al. [109], reported that eight out of eleven (73%) patients with relapsed or refractory CD19-positive cancers (non-Hodgkin's lymphoma or chronic lymphocytic leukemia [CLL]) had a response to anti-CD19 CAR NK without the development of major toxic effects. Also, these infused CAR-NK cells expanded and persisted in patients at low levels for at least 12 months [109].

For the neuroblastoma treatments, NK cells might requisite proper CARs constructs such as costimulator and/or signaling molecule, which are discussed in the next topic. Furthermore, several researchers informed that antitumor properties of CARs NK cells were downregulated by suppressive molecules of the TME, such as TGFβ, thereby limiting the tumor-killing capability of NK cells [99,110]. In this direction, a combination with the inhibitor of immunosuppressive molecule (e.g., anti-TGFβ) might increase the antitumor activities of neuroblastoma-targeted CARs NK.

3.1.3. NKT cells

NKT cellsare a subset of innate lymphocytes that share T and NK cells properties [111,112]. NKTs have differences in TCR diversity and their cancer immune response, and can be divided into two subtypes; invariance (type 1, iNKT) and variance (type 2) NKT cells [103]. NKT type I enhance antitumor immunity, while NKT type II regulate and inhibit tumor immunity [113-115]. iNKT cells have been studied and applied as potent tumor-specific CAR effector against lymphomas [93,116]melanomas [117], and neuroblastomas [86]. The first use of CARs NKT cells against neuroblastoma had been reported since 2014 [86]. Both 2ndand 3rdgeneration of anti-GD2 CAR NKT cells exhibited potent antitumor activity and significantly presented prolong survival without any graft-versus-host disease in a mouse xenograft model [86]. Unlike T cells, NKT effectively traffic and infiltrate into the tumor site [93]to mediate antitumor activities, inhibit tumor-supportive macrophage, and transactivate the localized NK and CD8+T cells [118,119]. Additionally, a high percentage of NKT cells in tumor-infiltrating lymphocytes have been detected in neuroblastoma than in patients' peripheral blood [120]. Likewise, Xu X. et al. reported that CARs NKT cells enhanced CARs treatment efficacy through their persistence property and provided the high localization onto the tumor site without significant histopathological toxicity [52]. These suggested that NKT cells are easily infiltrated into neuroblastoma than T lymphocyte. Recently, Heczey A, et al., [121]reported interim results from three patients with relapsed or resistant neuroblastoma, who enrolled into the phase I dose escalation trial, receiving the dose level 1 of autologous NKT cells co-expressed GD2 CAR and interleukin-15 (NCT03294954). Results exhibited that GD2 CAR NKT cells expanded in vivo,localized to tumors, and in one patient, induced regression of bone metastatic lesions without adverse effects [121].

Neuroblastoma-targeted CARs NKT cells are noticeable as the potential effector cells against neuroblastoma and other solid tumors. Of note, an exhausting phenotype of NKTs upon in vitro stimulation is associated with the upregulation expression of PD-1 and TIM-3 [93]. Thus, CAR NKT cells should be used in combination with anti-PD-1 or anti-TIM-3 to generate an intense tumor-killing effect.

3.1.4. γδT cells

γδT cells are innate T lymphocytes that independently recognize MHC targets, have a natural killing activity against pathogens, and respond to the tumor cells [122-124]. In contrast to αβT cells, γδT lymphocytes are rare in the peripheral blood and lymphoid organs (< 10%), while they mostly resided in a wide variety of tissues including the skin epidermis, gastrointestinal tract mucosa, and the reproductive system[125,126]. Interestingly, γδT cells also found in neuroblastoma tumors [120]. γδT cells can be divided into three subpopulations depending on the differences in TCR-δ chain expression; Vδ1, Vδ2, and Vδ3, and paired with the γ9 chain [127]. γδT cells expressing the Vγ9Vδ2 TCR is a common subset that can inhibit proliferation, angiogenesis, and encourage tumor cell apoptosis [103]. Moreover, these γδT cells can be activated and proliferated ex vivofor more than 1000 fold expansion [128].

Neuroblastoma mostly escapes from the immune system via the downregulation of MHC class I molecule [120]. Since γδT TCR does not require MHC-mediated antigen presentation [129,130], γδT cells may be appropriate to address this issue. Furthermore, these cells can differentiate into professional antigen-presenting cells (pAPCs), which present the antigenic fragments for CD4+and CD8+αβT cells [131]. Although γδT cells exhaustion is associated with PD-1, an immune checkpoints molecule [132], γδT cells have lower expression of PD-1 compared to αβT cells, thereby recognized as a promising effector cell for CAR immunotherapy. Like T cells, the naïve and memory phenotype of γδT cells sufficiently survive to promote persistence of CAR γδT cells in the recipients without decreasing the [133]killing activity against tumor cells [122].

Until now, few studies of CARs γδT have been reported. The first anti-CD19 CAR γδT cells is showed strong killing efficacy against leukemia [123,133,134]. For solid tumors, there are preclinical studies of CARs γδT cells against melanoma [87]and neuroblastoma [128,135,136]. Therefore, γδT cell is an influential candidate for developing of CARs immunotherapy against neuroblastoma. At present, two phase I clinical trials are available in clincaltrial.gov. The first study (NCT04107142) is planning to investigate of the safety and tolerability of NKG2DL-targeting CAR γδT cells in various solid tumors; colorectal, breast, sarcoma, nasopharyngeal carcinoma, prostate, and gastric cancer. While the second clinical trial (NCT03885076) is estimating the potential role of anti-CD33-CD28 CAR γδT cells against acute myeloid leukemia (ALL).However, the both clinical study results are not yet available. From these properties, the γδT cell is a potential candidate for using as an effector immune cell in adoptive CARs immunotherapy against neuroblastoma.”

4. When introducing the combination of PD-1/PDL-1 and CARs, it is suggested that the author give some specific examples in the literature or in clinical experience of CAR-based neuroblastoma therapies.

RESPONSE: We agree to this Reviewer’s comment. As a result, the new texts described specific examples in the literature and in clinical experience of the combination of PD-1/PDL-1 and CARs were added as followed;

Page 13-14; line 468-492:

“… T cell activation [184]. By using immunohistochemical analysis, Zuo S et al., [185]  reported that up to one-third (11/31) of neuroblastoma patients showed a positive PD-L1 expression in tumor tissues. It was not surprising that the PD-L1 positivity was associated to a decreased overall survival [185]. PD-L1 expression in metastatic neuroblastomas also play a key role in immune resistance mechanisms [186]. Several reports suggested that inhibition of PD-1/PD-L1 interaction by anti-PD-1 or anti-PD-L1 antibody enhance T cell killing efficacy to tumor cells [67,187,188]. Li S et al reported that combining anti-PD-1 antibody with CAR T cells engineered to secrete PD-1 inhibitor can enhance expansion and inhibitory effects in a human lung carcinoma xenograft mouse model [189]. The modified human CAR T cell secreted PD-1 blocking scFvs (E27 secreting CAR T) had also been established and investigated for therapeutic efficacy in hematopoietic malignancies as compared to their prototype CAR T cells. The results showed that E27 secreting CAR T significantly eliminate tumor cell and increase survival of tumor-baring mice [190]. Rupp LJ, et al. successfully established PD-1 deficient anti-CD19 CAR T cells combined with Cas9 ribonucleoprotein (Cas9 RNP)-mediated gene editing to disturb PD-L1-positive tumor xenografts. Results showed that Cas9-edited CAR T cells can be improved the therapeutic efficacy for cancer immunotherapy in vivo [191]. Since PD-L1 is highly up-regulated in multiple solid tumors, the PD-L1-target CAR T cells has been designed. A vector baring the extracellular and transmembrane regions of human PD-1; dPD1z, and a CAR vector against PD-L1; CARPD-L1z, have been established. Both dPD1z and CARPD-L1z effectively suppress the growth of multiple types of tumors in patient-derived xenograft models [192].

The combination of anti-PD1 and CAR T therapy have been investigated in phase I clinical trials for hematological (NCT04134325, NCT03287817, NCT04337606) and malignant pleural mesothelioma (NCT04577326). Likewise, the safety and efficacy of anti-PD-L1 combination with CAR T cells have been investigated for B-cell malignancies (NCT03310619) and various type of solid tumor including cervical cancer, sarcoma, non-small cell lung cancer (NCT04556669). In this direction, checkpoint inhibition may be turning neuroblastoma microenvironment…”

5. It is suggested that the author integrate the “Future perspective” of the fourth part and the “Conclusions” of the fifth part into one part.

RESPONSE: This is a good suggestion. The “Future perspective” of the fourth part and the “Conclusions” of the fifth part were then integrated into the last part of the manuscript Conclusion remarks and future perspectiveson page 14-15.

6. Some typos needed to be corrected, eg: “… This 1st generation anti-L1-CAM CAR efficacy and safety was investigated in patients with relapsed/refractory neuroblastoma in phase 1 clinical trial”, “It is known that there are challenges in the way of choosing an optimal CAR T cell target antigen in neuroblastoma is challenging …”, “2.3.4. CAR trafficking”.

RESPONSE:  We apologize for these typos. As the Reviewer’s guidance, the typos were corrected as following;

Page 4; line 123:

“This 1stgeneration anti-L1-CAM CAR efficacy and safety were investigated in patients with relapsed/refractory neuroblastoma in phase 1 clinical trial [12].”

Page 7; line 163:

“It is known that there are challengesin the way of choosing an optimal CAR T cell target antigen in neuroblastoma is challenging since many target antigens are related to normal peripheral nerves or neural tissue expression.”

Page 7; line 190:

“2.2.4. CAR trafficking”

7. The reference format needs to be unified according to the requirements of the journal.

RESPONSE: We thank the Reviewer for this comment. As a result, the reference format was unified according to the requirements of the journal.

Reviewer 2 Report

This is a well-described review to the current state of the art and future advancements of CAR-T therapy in neuroblastoma. This review will be a good introduction to the domain for non-experts.  

However, while the manuscript is globally understandable, the English needs to be improved. 

Author Response

Reviewer#2

This is a well-described review to the current state of the art and future advancements of CAR-T therapy in neuroblastoma. This review will be a good introduction to the domain for non-experts. However, while the manuscript is globally understandable, the English needs to be improved. 

RESPONSE: The authors are pleased that the Reviewer found this manuscript was well-written and globally understandable. The manuscript was edited to improve the English as the Reviewer’s suggestion, while all changes were labeled in red texts.

Reviewer 3 Report

The authors of the article “Strategies to improve chimeric antigen receptors 3 therapies for neuroblastoma” reviewed chimeric antigen receptors (CARS) and their use in neuroblastoma, a common pediatric solid tumor. The authors reviewed recent interventions in CARs therapy, including among others, multiple approaches in genetic engineering of T cells, and strategies to overcome T cell persistence, exhaustion, off target toxicity, and tumor microenvironment. Except for minor typos, the review is well written, comprehensive, up to date, an emerging new approach, and an interesting topic in cancer research. The proposed summary of the strategies in Fig. 3 is an excellent addition to the review.

It should be noted that a similar review, including similar details has been published in 2018 by Richards, MR et al. Hence, the way this review article was written does not bring additional insights into CAR T cell therapy of neuroblastoma. I would have expected that the authors, in addition to referencing research work by others, they would clearly highlight the contribution of their own research work, and what new information it brings in this field. Their research work is buried in the text without ability to differentiate it from others. Except for few other details, this review article brings little update of what is already reviewed in neuroblastoma therapy. Should the authors decide to write the review with this in mind, i.e. clearly explaining their own contribution in this topic, and how similar or different from others, it may become better suited for publication.

Minor corrections of the text:

Line 41- patients with relapsed-should be relapse

Line 98 last words: "has been being". delete being

Line 115: "have been reports". correct to have been reported

Line 145/146: delete is challenging

line 153: "in neuroblastoma studying" should be studies

Fig. 3: the authors are encouraged to include on-target off-tumor toxicities.

Line 234: "the memory subtype of CAR T cells is exigency". Should be an exigency.

Line 356: this paragraph should be with section 2.2.3

Author Response

Reviewer#3

The authors of the article “Strategies to improve chimeric antigen receptors 3 therapies for neuroblastoma” reviewed chimeric antigen receptors (CARS) and their use in neuroblastoma, a common pediatric solid tumor. The authors reviewed recent interventions in CARs therapy, including among others, multiple approaches in genetic engineering of T cells, and strategies to overcome T cell persistence, exhaustion, off target toxicity, and tumor microenvironment. Except for minor typos, the review is well written, comprehensive, up to date, an emerging new approach, and an interesting topic in cancer research. The proposed summary of the strategies in Fig. 3 is an excellent addition to the review.

The authors appreciate that the Reviewer find this review is well-written, comprehensive and up to date. Please find below details of how all comments have been fully addressed in the manuscript.

It should be noted that a similar review, including similar details has been published in 2018 by Richards, MR et al. Hence, the way this review article was written does not bring additional insights into CAR T cell therapy of neuroblastoma. I would have expected that the authors, in addition to referencing research work by others, they would clearly highlight the contribution of their own research work, and what new information it brings in this field. Their research work is buried in the text without ability to differentiate it from others. Except for few other details, this review article brings little update of what is already reviewed in neuroblastoma therapy. Should the authors decide to write the review with this in mind, i.e. clearly explaining their own contribution in this topic, and how similar or different from others, it may become better suited for publication.

RESPONSE: This is an important comment. We have worked on anti-GD2 CAR T cells for neuroblastoma during the last few years and have encountered obstacles in the process that led us to look for possible solutions. With that goal in mind, we wrote this manuscript to serve our research and benefit other researchers who plan to conduct, or currently working on, CAR therapy for neuroblastoma. We have already noticed that the review published in 2018 by Richards, MR et al. provides excellent information on CAR T cell therapy in neuroblastoma and their significant contribution works. We would like to emphasize that this revised manuscript provides some more updated information (e.g., clinical trials registered during 2018-2020, effector cells, combination therapy) that are complementary to Richards MR, et al. and can be listed as follows; 

  • Up to date summary of clinical trials (please see the revised Table 1)
  • Up to date target antigens studies both in preclinical and clinical studies.
  • Information regarding CAR strategies that are not redundant to Richards MR, et al.;
    • Candidate effector cells
      • NK (up to date information)
      • NKT
      • γδT
    • CAR structure
      • The suitable or mostly used spacer for each candidate effectors
      • Co-stimulatory domain for each candidate effectors
      • Signaling domain: CD3ζ vs. DAP12
    • TME
      • Enhance T cell trafficking 
      • Evidence in CXCR2 CAR, using IL-8, or any inhibitors of immunosuppressive molecules
    • MDSC
      • Combination of NKD2G NK cell and CAR T cell for enhancing CAR T in solid tumor 
    • Tumor extracellular vesicles
    • Combination CAR with;
      • PD-1/PD-L1 inhibitor
      • IL-15 cytokines expression

We hope the Reviewer would comprehend our situation and find the revised manuscript becoming better suited for publication.

Minor corrections of the text:

Line 41- patients with relapsed-should be relapse

RESPONSE: This typo was corrected already;

Page 1; line 41:

“… patients with relapse [7,8].”

Line 98 last words: "has been being". delete being

RESPONSE: This error was corrected as suggestion;

Page 4; line 114:

“… the efficacy of anti-B7H3 CAR has been being demonstrated in vivo

Line 115: "have been reports". correct to have been reported

RESPONSE: This typo was corrected as following;

Page 4; line 131:

“Many approaches of 1stgeneration of anti-GD2 CAR have been reported...”

Line 145/146: delete is challenging

RESPONSE: This error was corrected as following;

Page 7; line 163:

“It is known that there are challengesin the way of choosing an optimal CAR T cell target antigen in neuroblastoma is challenging since many target antigens are related to normal peripheral nerves or neural tissue expression.”

line 153: "in neuroblastoma studying" should be studies

RESPONSE: This typo was corrected already;

Page 7; line 171:

“… in neuroblastoma studies

Fig. 3: the authors are encouraged to include on-target off-tumor toxicities.

RESPONSE:We appreciate this Reviewer’s suggestion. However, Figure 3 was designed to enhance the understandability of the section 3 (strategies in improve CARs in neuroblastoma). Since the concept of on-target off-tumor effect has already been provided in the section 2.2.2., so we believe Figure 3 as its current form serves our intention. We hope the Reviewer will comprehend our decision.

Line 252: "the memory subtype of CAR T cells is exigency". Should be an exigency.

RESPONSE: The article ‘an’ was added as the reviewer’s suggestion;

Page 9; line 251:

“… the memory subtype of CAR T cells is an exigency”

Line 356: this paragraph should be with section 2.2.3

RESPONSE:We apologize for this confusion. The section 3.3. want to deliver information regarding the strategies to overcome immunosuppressive TME in order to improve the efficacy of CAR immunotherapy. Therefore, the section title was changed to prevent this misunderstanding as following;

Page 12; line 420:

3.3. Overcoming TME to improve the efficacy of CAR immunotherapy

Some redundant information in the section 3.3.1. were removed as following;

Page 12; line 427:

“The low T cell infiltration has been observed in neuroblastoma so-called immunologically “cold” tumors [125]. Various studies demonstrated that poor trafficking and limited CAR T cell persistence to solid tumors are significant burdens in CAR immunotherapy [126]. Tumor chemokines and their receptor play an essential role in TME [127]. Chemokines also mediate proliferation, infiltration, and persistence of leukocytes [128], which affected tumor control [129]. Aberrant expression of tumor chemokine-chemokine receptor network affected several tumor biological functions, i.e., stimulating tumor cell proliferation, inducing tumor angiogenesis [130], enrolling inhibitory immune cells [131], preventing recruitment of effector lymphocytes [132], and resisting their antitumor response [133], leading to cancer immune escape [134].To increase trafficking ability of …”

Round 2

Reviewer 3 Report

The manuscript has been revised to include latest trial updates, and additional information added that reflect latest research in CAR therapy.